# Understanding the Problem of Access to Public Health Insurance Schemes among Cross-Border Migrants in Thailand through Systems Thinking

**DOI:** 10.3390/ijerph17145113

**Published:** 2020-07-15

**Authors:** Watinee Kunpeuk, Pard Teekasap, Hathairat Kosiyaporn, Sataporn Julchoo, Mathudara Phaiyarom, Pigunkaew Sinam, Nareerut Pudpong, Rapeepong Suphanchaimat

**Affiliations:** 1International Health Policy Program, Ministry of Public Health, Tiwanon Road, Nonthaburi 11000, Thailand; hathairat@ihpp.thaigov.net (H.K.); sataporn@ihpp.thaigov.net (S.J.); mathudara@ihpp.thaigov.net (M.P.); pigunkaew@ihpp.thaigov.net (P.S.); nareerut@ihpp.thaigov.net (N.P.); rapeepong@ihpp.thaigov.net (R.S.); 2Faculty of Business Administration and Technology, Stamford International University, Motorway Road—Km2, Prawet, Bangkok 10250, Thailand; pteekasap@gmail.com; 3Division of Epidemiology, Department of Disease Control, Ministry of Public Health, Nonthaburi 11000, Thailand

**Keywords:** systems thinking, migrants, health insurance, health policy

## Abstract

Thailand has become a popular destination for international migrant workers, particularly from Cambodia, Lao PDR, and Myanmar. However, only a fraction of these migrant workers were insured by public health insurance. The objective of this study was to apply systems thinking to explore contextual factors affecting access to public health insurance among cross-border migrants in Thailand. A group model building approach was applied. Participants (*n* = 20) were encouraged to share ideas about underlying drivers and barriers of migrants’ access to health insurance. The causal loop diagram and stock and flow diagram were synthesised to identify the dynamics of access to migrant health insurance. Results showed that nationality verification is an important mechanism to deal with the precarious citizenship status of undocumented migrants. However, some migrants are still left uninsured. The likely explanations are the semi-voluntary nature of the Health Insurance Card Scheme, administrative delay of the enrollment process, and resistance of some employers to hiring migrants. As a result, findings suggest that effective communication is required to raise acceptance towards insurance among migrants and their employers. A participatory public policy process is needed to create a good balance of migrant policies among diverse authorities.

## 1. Introduction

Globally, in 2018, there were about 258 million international migrants and approximately 60% were migrant workers who travelled abroad to address labour shortages in destination countries [1,2]. The rise of cross-border migrants worldwide attracted political attention as reflected in the number of high-level dialogues and commitments. For example, the 2030 Sustainable Development Goals emphasised the inclusiveness of health among all populations, leaving no one behind [3]. The health of migrants and refugees was specifically raised in the 2017 World Health Assembly (WHA) Resolution 70.15, ‘Promoting the Health of Refugees and Migrants’ [4]. According to this Resolution, Member States are urged to strengthen international cooperation regarding the health of refugees and migrants [5]. The provision of necessary health-related assistance through bilateral and international cooperation was recommended for countries with a high influx of refugees and migrants. In addition, a situation analysis about best practices and lessons learned in various contexts was requested as part of the Global Action Plan on the health of refugees and migrants stipulated in the seventy-second WHA in 2019 [4].

Thailand has been a popular destination for migrant workers in the Southeast Asia Region for many years. The majority of immigrants travel from neighbouring countries, especially Cambodia, Lao PDR, and Myanmar (CLM) [6]. In 2018, the number of migrants in Thailand was approximately three to four million, and about one and a half million crossed the border without a valid passport or travel pass and were recognised as undocumented migrants [7]. As there is a concern about national security, the Thai government established the One Stop Service (OSS) and the Management Center for Migrant Workers (MCMW) as a mechanism to legalise undocumented migrant workers. The role of both authorities is to record personal data, coordinate with the health sector for health examinations, and work with the Ministry of Interior (MOI) to issue legitimate residence permits and cooperate with the Ministry of Labour (MOL) to issue work permits. All of these functions are known as the nationality verification (NV). In terms of health protection, public health insurance for migrants is one of the most important areas of social policies in Thailand. Overall, the Thai government has implemented two main insurance schemes. The first is the Social Security Scheme (SSS) which covers formal sector workers regardless of their nationality (Thai nationals in the formal sector are also enrolled in the SSS). Migrant workers in formal sector, in theory, need to be insured; the SSS is a compulsory insurance scheme that regulates employers to contribute to a payroll tax, which is equally shared by their employees. The scheme is financed by tripartite contributions. Migrant workers need to pay 5% of their income to the SSS fund, with employers’ subsidies at 5% and the Thai government’s at 2.75% [8]. The second scheme is the Health Insurance Card Scheme (HICS), regulated by the Division of Health Economics and Health Security at the Thai Ministry of Public Health (MOPH). The HICS covers migrant workers in the informal sector from CLM nations [9] (see Figure 1).

Despite the existence of the NV and public health insurance schemes for migrants, challenges remain. For instance, the MOL reported in 2018 that about 800,000 out of 2,000,000 migrants registered for the NV could not complete the process within the specified timeframe [10]. In reality, there remain migrants (an unknown figure) that have not joined the NV as intended by the government. Suphanchaimat et al. [9] describe these challenges in relation to numerous factors such as ignorance of policies among migrants and employers, a swift change of policies due to both international and domestic pressures, and incoherence of policy direction among concerned authorities. However, no work has used a system science approach to assess the structural problems that limit migrants’ access to public health insurance. In the public health field, systems science methods, particularly systems thinking and system dynamic modelling, have been widely used to map components of health systems; examine and compare the potential outcomes of health interventions to guide more efficient investments; and enhance the process of policy decision-making [11,12]. Advantages of systems thinking are reported as it can suggest a range of health policy strategies to cope with different populations and conditions [11]. For researching the experiences of migrant populations, systems thinking and systems dynamic modelling have been applied to various areas, but have not yet been applied to the issue of health insurance. For instance, Pedamallu et al. [13] employed a system dynamics model to identify factors influencing academic performance among migrant students in Turkey. To assess economic impact, a dynamic simulation model was used to estimate the migration flows and labour markets between countries for economic systems development [14]. In the context of Southeast Asian countries, systems thinking was employed in a qualitative study to explore stakeholder perceptions towards challenges in migrants’ and health workers’ language and cultural competency in Thailand and Malaysia [15]. To unpack these challenges, a thorough understanding of all migrant health policies (including the NV process, the issuance of work permits and insurance measures) and how these policies interact with each other is necessary. This point is the main objective of this study. To address this, we opted to apply a systems science approach, which is a useful technique to help illuminate the function of and the interaction of components in a complex system based on mutual engagement of relevant stakeholders [16]. It should be noted that the main focus of this study was migrant enrollment in public health insurance schemes (SSS and HICS).

## 2. Materials and Methods

### 2.1. Study Design and Data Collection

This study applied a qualitative approach based on the concepts of systems dynamics and group model building (GMB) [17,18]. Due to the complex characteristics of health policies and systems, the application of system science is considered helpful to identify how the system works, what the main problems are, and how to sustain the systems with minimal resistance from all concerned parties [16]. The GMB is a participatory process and is part of systems thinking that engages diverse key informants to provide insightful comments in a model analysis [18,19]. It allowed stakeholders to share their mental models on the topic [20,21], which can minimise bias from researchers. Two rounds of GMB workshops lasted 150–180 min and were organised at the office of the International Health Policy Program (IHPP), MOPH. The GMB facilitation team comprised three senior modelers, one junior modeler, two gatekeepers, and two rapporteurs. Most of the facilitators were IHPP academic staff; only one facilitator was an independent academic expert invited from outside IHPP.

Participants in the GMBs were purposively selected. More specifically, diverse groups of health staff, frontline health professionals, representatives from the MOPH, and policy makers were included in the GMBs. Experts from the MOL, the MOI, and Non-Government Organisations (NGOs) also participated.

### 2.2. The GMB Process

The GMB started with an overview presentation of the migrant health situation in Thailand with a focus on the migrant journey towards obtaining health insurance. The scope of discussion was limited to low-skilled workers from CLM countries. Highly skilled professionals, expatriates, tourists, asylum seekers and refugees were excluded. The discussion then delved into the two main insurance arrangements of the SSS and HICS. Two GMBs ran separately on 24 July 2019 (14 participants) and 29August 2019 (10 participants). Some participants joined both GMBs and therefore the total number of individual participants in both GMBs was 20. The additional information of participants was described in the Appendix A. At the end of each session, participants were asked to identify leverage points that critically determined migrant access to health insurance. Leverages are considered to be any factor where its change can have a substantial impact on the whole system [22]. All discussions were led and run by the participants with minimal interference from the facilitators. The description of participants’ profiles is presented later in the results section.

### 2.3. Causal Loop Diagramme (CLD) and Stock and Flow Diagramme (SFD)

The participants were asked to share ideas of the draft CLD. The CLD demonstrates associations among factors relating to insurance enrollment in a cause-and-effect manner. The facilitators drew the CLD on a flipchart based on input from the participants. The research team explained the draft CLD to participants using lay language, and then asked the participants to specify whether a concerned variable exhibited positive (+) or negative (−) influences on other variables. Another key feature in the CLD is a feedback loop which occurs when the effects of consequences have returned to influence the causes [23]. There are two types of feedback loops: reinforcing and balancing. Reinforcing loops indicate that influences of variables in the same loop are travelling in the same direction, whereas balancing loops show the travel in opposite directions. The authors also developed SFD to identify the dynamics of the HICS enrollment. The SFD is part of system dynamics, which consists of stocks (the rectangular shapes) and flows (the thick arrows). The stocks indicate accumulation of anything which can be quantified and changed over time. The flows show the direction of change occurring from any stocks which can be inflows or outflows [24]. Then, the research team translated the CLD and SFD from the flipchart to an electronic format by Vensim software [25].

### 2.4. Data Management and Analysis

Data were analysed by both inductive and deductive thematic coding. All of the discussions and interviews were transcribed verbatim and then manually coded in an Excel programme. Deductive coding was used at the beginning to develop the question guide for facilitators during the group process. Then open coding was performed based on inductive thematic analysis to discover common concepts from the discussion. After that, axial coding was employed to categorise the data into main groups by the research team. In each main group, subcategories were identified by using selective coding. Analytical memos were created to map the categorised data with reviewed literature for triangulation process. The interviewees were coded as letters to maintain anonymity. The products of thematic analysis were finalised and validated by feedbacks from experts in the field and stakeholders involving in both rounds of GMB. The flowchart of data collection and analysis is shown in Figure 2. After this analysis, insights from emerging themes were again mapped and synthesised in the form of CLD and SFD. Concerning data security, only the main researcher (WK) could access to the data. The collected data in this study will be deleted within two years after the completion of the project.

### 2.5. Ethics

Ethics approval was obtained from the Institute for the Development of Human Research Protections in Thailand (IHRP 340/2562). All participants were asked to sign a consent form. The research team assured the participants that their practices were on a voluntary basis and if they felt uncomfortable about participating in the discussions or interviews, they could drop out these activities. Anonymity was protected by coding individual identification numbers instead of noting individual names.

## 3. Results

### 3.1. Characteristics of Stakeholders

The majority of stakeholders were aged 40 years and older. One third of participants had been working for more than 30 years. Over half of the participants were at senior professional level. About half of the participants were from the health sector. Approximately 40% of the attendees were from civic organisations (Table 1).

### 3.2. Key Important Themes Identified From the GBMs

Five themes emerged from the GMBs: (i) NV process as the most critical step for legalising the precarious status of undocumented migrants; (ii) Role of private hospitals in health examination before enrolling in the insurance; (iii) Interim period between enrolLment in the SSS and the effective activation of insurance; (iv) Practical problems originating from the difference in design between the SSS and the HICS; and (v) Data recording system of the HICS. The SFD (Figure 3) combined key components in the loop of legalisation process and access to the health insurance schemes (SSS and HICS) with an integration of the five subthemes. The CLD depicted key variables in the closed loop, particularly in access to the HICS (Figure 4).

#### 3.2.1. NV Process as the Most Critical Step for Legalising the Precarious Status of Undocumented Migrants

Migrants with illegal status might opt to enter Thailand first then undertook the NV later in order to obtain a legal work permit (as shown in the cloud symbol on the left-hand side of Figure 3). The NV and the issuance of a work permit served as a compromised measure that accounted to balance the need of national security and economic demand. For addressing the illegal status of migrants, some participants (SD6 and SD15) referred to the 20-year National Strategy Plan (2018–2037), which stated that illegal migrants must be deported [26]. However, in reality, this measure was not often exercised as the government needed to take into account the balance of the demand for labour, which is driven by economic necessity.

The dynamics of migrants in the labour market was remarkable. Not all registered migrants were in the labour market all the time. Some might leave the labour system due to personal difficulties, work injuries, or financial hardship [27]. As the minimum wage in Thailand was higher than that in neighbouring countries, Thailand was an attractive destination providing better job prospects for cross-border migrants. From 1 January 2020, any workers in the central area were paid the minimum wage of THB 331 (USD 10.7) regardless of their nationality [28]. However, some participants (SD7, SD14 and SD15) mentioned that the process to enter the country in a lawful manner was complicated and migrants with low educational backgrounds or poor economic status found it difficult. As such, some undocumented migrants always avoid lawful border-crossing mechanisms.

From the participants’ point of view, the NV process was the most critical step to determine whether an undocumented migrant would be insured as part of the government’s registration process, the OSS. The OSS was the official mechanism based on multi-sectoral collaboration across five ministries: (i) Department of Provincial Administration, the MOI of Thailand; (ii) Department of Employment, the MOL of Thailand; (iii) Immigration Bureau, the MOI of Thailand; (iv) the Social security Office (SSO), the MOL of Thailand; and (v) the MOPH of Thailand. Once registered, the Provincial Administration Office collected migrants’ personal records including fingerprints. Then the registered migrant was given an identity card with a unique identity number printed on it (13-digit code). The Department of Employment then issued a work permit once the 13-digit code was confirmed. The registered migrants obtained a temporary passport and work visa by the Immigration Bureau. The alternative national verification office was the MCMW [10], located along the border of the migrants’ countries of origin. The mission of the MCMW was mandated to coordinate with the Provincial Labour Office or the Bangkok Labour Office to facilitate the NV.

Migrants who passed the NV would be eligible to lawfully stay in Thailand for a certain time period (normally two years). The proof of evidence of completing the NV was a temporary passport with visa. If a migrant did not completely pass the NV, he or she would only receive the pink card as evidence of entering the legalisation process. Some participants reported that there were some misunderstandings about the policy and not all government officers were knowledgeable about the NV process. Some officials considered migrants holding the pink card as an illegal person. This factor partly prevented migrants from accessing social security benefits, including health insurance. This theme was shown in S1 (Figure 3).

“After the NV, migrants are legalised as they own a passport and a pink card. Information about each person is later identified in the civil registration section (CVS). While they are waiting for their name to be listed in the CVS, some government officers think that they are still illegal. And they also think that having a pink card refers to illegal status.”—Male participant, SD7 NGO.

#### 3.2.2. Role of Private Hospitals in Health Examination before Enrolling in the Insurance

Health examination was one of critical steps for legalising migrants’ status before being insured and also positively impacts public health security. However, the unclear policy message from the central MOPH about the validity of health examination resulted from private hospitals was reflected by the interviewees (S2 in Figure 3). The participants said that, in the field, there were conflicting ideas as to whether and to what extent private facilities could take part in the health check for migrants before these migrants were enrolled in the insurance. This is because one of the pre-conditions for a migrant to obtain a work permit was to pass a health check, which screened for serious communicable diseases such as tuberculosis, filariasis and elephantitis [29]. The problem raised was that the MOL regulation allowed either a private hospital or public hospital (with certified standards as approved by the Hospital Accreditation System, or the Joint Commission International or International Standardization Organization) to be responsible for the health check. However, almost all the contracted facilities of the HICS and the SSS were state run and the MOPH regulation did not specify that the hospital performing the health check must be the same hospital that provided the HICS. This approach created some problems as the majority of private health facilities do not sell health insurance. Therefore, migrants gained the approval of the health check from such facilities but most of them did not continue buying HICS, which was normally sold by public hospitals elsewhere in the area.

Moreover, some MOL local officers denied the result of the health check, particularly from private hospitals, as mentioned by the respondents.

“The problem is that migrants just went to private hospitals for their health check approval. Many of them do not go further to buy HICS at public hospitals near their workplace. Moreover, some providers perceived that the health check result from a private hospital is invalid (and did not allow migrants to buy the HICS further).”—Female participant, SD 4 public health officer.

Some participants stated that some provinces established an internal agreement among the health facilities within the province that only migrants passing the health exam from public facilities would be eligible to purchase the HICS. Some solutions were suggested by the participants. One participant suggested that the HICS regulation should be amended and this necessitated a strong legal support.

“We communicated within our province to accept only those health check results from public hospitals and declined others from private hospitals or clinics.”—Female participant, SD 3 public health officer.

“We need a legal approach to tackle this challenge by revising the law and we need to propose this issue to the Cabinet. The Royal Decree for Health Examination and Health Insurance for Obtaining Work Permit at the status quo (that allows private hospitals to make out a medical certificate) should be cancelled.”—Male participant, SD 6 NGO.

#### 3.2.3. Interim Period between Enrollment in the SSS and Effective Activation of Insurance

Types of migrants’ employment defined the health insurance schemes that migrants were insured. Migrants in formal sector were covered by the SSS, whereas all migrant populations excluded by the SSS were eligible for the HICS (see Figure 3). A key problem lied in the 90-day interim period before the insurance of migrants became activated. The migrants’ employers and the migrants were obliged to contribute to the payroll for 90 days until the right to the SSS insurance was activated, and the HICS was an option for insuring migrants at this stage. Although in principle the HICS is set for informal sector migrants, in practice the purchase of the HICS is not strict and varies across hospitals. The MOPH has initiated many subtypes of HICS with varying periods of coverage (for example six-month, one-year, and two-year HICS), Table 2 [9]. About half of the participants emphasised that local health staff that tended to sell the six-month HICS or the one-year HICS all migrants despite the fact that some migrants were facing the 90-day SSS-free period. This practice happened because some hospitals tended to sell the HICS with extended coverage to create a greater pool of revenue. In some circumstances, health staff became active agents in convincing employers to encourage HICS enrollment. These problems were reflected in S3 (Figure 3).

“The six-month or one-year HICS is more favourable than the three-month-HICS as practically a small number of migrant workers will use this right over the 3-month period. The six-month or one-year HICS is a better option because of the longer insured period. Sometimes employers thought that it was not their business (to buy the insurance for migrants) but the hospital insists (that migrants need to be insured by the HICS). The SSO also is not sure if after the three months the employers follow up the SSS for their employees again. It seems this mission (insuring) is not the responsibility of other ministries but of the MOPH.”—Female participant, SD4 public health officer.

#### 3.2.4. Practical Problems Originating from the Difference in the Design between the SSS and the HICS

Most participants pointed to the problems involved with the difference in the payment methods between the SSS and the HICS. One of the common concerns was that some employers and migrant employees were not willing to have their salary deducted as a payroll contribution. In contrast, the purchase of the HICS was a lump sum payment once a year (or in two years according to HICS subtypes). Another problem raised was the legal basis of the HICS and the SSS. The SSS was established according to the Social Security Act 2010 [30], while the HICS was based on the Ministerial Announcement. The Act had a greater hierarchy within the law with a penalty specified for those breaching the law but this did not apply to the Announcement.

“Some migrants declined to buy HICS once they came into Thailand. This is because there is no mandate that forces migrants to buy HICS”—Male participant, SD 5 NGO.

This problem was aggravated during transition of job types when a migrant changed jobs from the formal to informal sector. Participants also mentioned that apart from the health insurance premium, the fear of being penalised was among key reasons that made migrants avoid government registration (which then resulted in missed opportunities to be insured with either the SSS or the HICS). These challenges were reflected via a symbol, S4, in Figure 3.

“If they have already paid for the SSS subsidy and if they later leave the job then they have to go for HICS.”—Male participant, SD 5 NGO.

“The SSS has advantage (over the HICS) because of the existence of the supporting law. If the employers do not follow this rule, there is no doubt that they will face a penalty. However, practically, some employers avoid this rule and leave their workers uninsured (either with the SSS or the HICS).”—Male participant, SD 6 NGO.

#### 3.2.5. Data Recording System of the HICS

Most participants mentioned the poor management of the HICS reporting system. The MOPH did not report publicly how many migrants were insured with the HICS or the breakdown of the characteristics of the insurees (for example, in terms of sex, age, and occupation).

“The problem is that the number of HICS sales should be publicly reported in the database of the Division of Economics and Health Financing, Ministry of Public Health. But now it seems we can’t track which group buys HICS the most and how many HICS are sold. This results in difficulty to monitor and evaluate HICS progress.”—Male participant, SD 6 NGO.

Some informants mentioned that health facilities were hesitant to report exact the HICS revenues to the MOPH. According to the regulation on HICS financing, part of the HICS revenues should be pooled at the MOPH for more equitable allocation of funding back to local health facilities upon requests for the reimbursement for high-cost treatment [9]. However, in practice, some health facilities ignored this measure as they were willing to bear the risk of high-cost treatment themselves. This situation was more pronounced in health facilities with a high density of migrant workers, while in some settings with a small number of migrants health facilities are more likely to report this number to minimise financial risk when facing high-cost patients.

“The MOPH can’t exactly analyse the number of HICS sales as they don’t know how to track this number accurately. Some hospitals feel uncomfortable with reporting HICS revenue to the MOPH as they are willing to take the risk of treatment cost for migrants. If you have about 100 migrant patients, I suppose you tend to report the revenue of HICS to the central MOPH, as it just a small number. In contrast, if you earn 5–7 million Baht from HICS, you won’t give this number to the MOPH because if you spend only 700 thousand Baht on treatment, the rest becomes your profit.”—Female participant, SD 4 Public health officer.

In summary all subthemes and key findings were briefly described in Table 3.

Key variables in access to HICS were captured in Figure 4. As the HICS does not have any penalty, participants agreed that the enrollment of migrant workers in the HICS depended on the discretion of each individual migrant to purchase the insurance (demand side) plus the willingness of health care providers to sell the HICS (supply side). On the demand side, the likelihood of migrants buying HICS is shaped by affordability of the insurance, knowledge and awareness about health rights among migrants, and their own health status. Migrants who have not developed any serious diseases are less likely to understand the benefits of being insured [31,32]. These findings were supported by previous literature [33,34], which indicated this as one of barriers of migrants’ access to public health insurance in Thailand. One participant mentioned that, in some provinces, there was a training course for migrants about basic rights and benefits of the insurance. Thus, the perception of human rights and knowledge of health insurance for migrant among employers in those provinces may also influence HICS enrollment.

The legal status of a migrant also influenced access to health insurance. This was because if those with illegal or undocumented status fail to register with the OSS, they were likely to keep themselves uninsured due to fear of deportation if they happened to expose themselves to authorities. One stakeholder informed that practically the system cannot deal with all illegal migrants. Many of them illegally stayed in Thailand with fear of the government penalty.

“The MOPH urges me to report the number of undocumented migrants in my area. But we can’t do it. We are willing to sell the HICS, even though they don’t have any personal identity evidence either a passport or pink card. And even they would like to buy it, fear of penalty discourages their attempt and so they won’t visit us.”—Female participant, SD4 Public health officer.

Regarding the economic aspect, the likelihood of the HICS purchasing was negatively associated with the HICS premium as shown in the balancing loop (B1) in the CLD (Figure 4). Access to HICS meant that insured migrants had less out-of-pocket (OOP) payment for medical expenses than uninsured migrants. In addition, the fixed premium of the HICS could serve as a barrier to HICS access among the poor and among the juveniles. This was because migrant dependents aged 7–18 years needed to pay HICS premium at the same price as migrant adults.

“The HICS premium should be categorised more specifically by age groups. I think each group has no equal affordability for this current HICS premium especially children aged 7–18 years. The HICS price for them is the same for migrant adults. It is unfair.”—Male participant, SD 5 NGO.

From the supply-side perspective, the HICS was a revenue generator for a hospital, especially where the insurees were large in number. The revenues from selling HICS could help a hospital’s financial status and meant financial security of the central MOPH. This relationship could be seen in the reinforcing loop (R1). However, effective data collection and reporting of the HICS by the MOPH were challenging as this needed more engagement from both local health facilities and the central MOPH.

Furthermore, the linkage between HICS and public health security could be observed as an accelerator to the likelihood of the HICS selling. The concern over spreading certain infectious diseases from migrants to the wider Thai public (such as tuberculosis, HIV, syphilis and malaria) was a major factor which encouraged providers to sell the HICS [35,36,37,38,39,40]. As a result, the relationship in access to the HICS and public health security was positive as shown in R2 in the CLD. In addition, NGOs and civil society from domestic and international organisations also played an active role in coordinating with government departments, employers, and employees to build trust and advocate social programmes related to human right and health in migrant communities [33].

## 4. Discussion

### 4.1. Result Discussion

Overall, this study employed a qualitative analysis with a concept of systems thinking to explore the mechanism and contextual factors involving in migrant health policies, ranging from the NV process to insurance measures. This study also demonstrated interactions of key elements in the system of migrant health policies that determine access to public health insurance in low-skilled migrant workers in Thailand. Five subthemes emerged and then served as inputs for developing CLD and SFD in the later part of the study. It is clear that the high labour demand, particularly in ‘3D’ jobs (dangerous, dirty, and demeaning), to address economic necessity is the main factor that drives a high influx of migrant workers [9,27,29]. Therefore, the private sector has a dominant role in maintaining the labour flow and has a strong power influencing the labour demand [33,41].

The majority of the participants concurred that the NV and the issuance of work permits are the most important step that lead to the uptake of insurance. In practice, the NV implementation is not perfect. The main challenge is the complexity and delays of the NV process. Findings suggested that this factor significantly impacts the decision of both migrant themselves and employers to avoid the NV and continue illegal activities. This structural barrier is mainly caused by incoherence of inter-sectoral policies and poor coordination across ministries [9]. To address this challenge, strict law enforcement is vital. Therefore, the government, in collaboration with all partners in society, should play an active role in encouraging all migrants to access the NV and obtain a work permit. To enhance multi-sectoral collaboration across the government departments, complexity in the NV process and bureaucracy needs to be minimised. Evidence suggests that a lack of integrative data collection system of migrant registry and other non-Thai populations; a resource- and time-consuming process of the NV; and the poor feedback report system of the government agency contribute to poor management of migrant registry and the NV [42]. Therefore, the responsive and supportive system should be integrated to engage both migrants and employers in the NV process; this measure can also deal with corruption and malpractices in the labour market. At local level, proactive actions across the MOI and MOL should be taken into account. A systematic and transparent monitoring system should be in place particularly in addressing employment without work permit [43]. To facilitate different needs from multi-sectoral partners, a central authority should be established with representatives from diverse professionals to coordinate with various sectors and enhance the NV implementation.

Apart from the government agency collaboration, the private sector and employers should be accountable to support transparent mechanism of the NV. This can be promoted by lawful migrant recruitment through a Memorandum of Understanding (MOU), which is expected to reduce interference by private intermediaries or brokers and prevent human trafficking [6]. Moreover, engaging employers and private sector in migrant policy process and decision making is needed. At national level, employers can have a role in determining demand and supply chain of migrant labour with the government partner. At individual level, private sector can involve in the government campaigns for promoting migrants’ awareness and knowledge on the right for health and social services, and acknowledging cultural diversity of migrant communities with different backgrounds [44].

The legal process to achieve the NV should not be limited to national security interest. Broader actions are needed to cover illegal border crossings. This requires a seamless international coordination across ministries, not only in Thailand, but also between Thailand and neighbouring countries [6]. One of the most recognised international collaboration in this region is the ASEAN Committee on the Implementation of the ASEAN Declaration on the Protection and Promotion of the Rights of Migrant Workers. This aims to safeguard migrant workers in different areas, including migrant worker management policies; migrant information; access to services and immigration requirements; and migrant overseas employment administration [45]. However, this is non-legally binding, which results in ineffective implementation in a real setting. Therefore, more legal mechanisms across countries are required to ensure a direct solution to improve migrant labour system in the ASEAN region [46].

Apart from individual factors from migrant themselves and employers influencing positive feedback in the demand side, economic reasons were critical that balanced the loop with negative direction. Although it is clear that HICS insurees tended to have less OOP payment than uninsured migrants [47], the HICS price was also significant to shape the migrants’ demand due to their differences in affordability. The high cost of fixed HICS premium is likely to discourage access to the HICS among the poor. Therefore, the revision of the HICS premium to cover all migrants with different economic statuses should be considered.

### 4.2. Methodological Discussion and Reflection of the Research Team

To the best of our knowledge, this work is the first study that applied a systems thinking approach to understand the dynamics of access to health insurance among migrants in Thailand. This approach is innovative in the Thai context to explore key variables influencing access to migrant health insurance by taking views of all concerned parties into account. The findings clearly prove that the existence of migrant insurance policy is not in itself a guarantee for access to insurance among migrants.

However, some limitations exist within the study. First, the process at this stage did not include migrants as participants. Further work that incorporates views of migrants as an insurance beneficiary will definitely provide additional insights. Second, there are groups of non-Thai populations unmentioned here including but not limited to, stateless people, expatriates and refugees. Therefore, interpretation of the findings to other non-Thai populations must be exercised with caution because each group of non-Thai populations has its own characteristics and face idiosyncratic challenges. Third, access to insurance does not necessarily mean access to care, let alone a better health outcome. Fourth, as the focus of this systems approach was based on public health field, some insights of political and economic factors might be absent from this analysis. However, the research attempted to minimise this bias by including participants working closely with the national security and economic sectors. Fifth, there are many factors that influence access to care and the quality of care received. These include adequate supply of health resources, the quality of the services provided, and the underlying health status of the clients (migrants), to name but a few. Last but not least, in terms of reflexivity, as the research team has been working closely with research and advocacy for better health access for migrants, it is likely that the researchers used humanitarian lens to interpret the data.

Concerning policy recommendations, to boost migrant access to the insurance, synergistic efforts are required from all stakeholders, including the government, employers of migrants, and migrant communities. Increasing awareness of the insurance among migrants and employers, strict law enforcement and seamless inter-sectoral policies are all needed. The gaps between HICS and SSS enrollment should be filled. The government should invest more in the NV process to make it more efficient, less time consuming and independent from the interference from private intermediaries. The purchase and selling of the HICS should be systematised and free from the discretion of an individual (either a migrant [client] or a provider [seller]).

Regarding further research priorities, there should be additional studies that explore the angles of healthcare access, quality of care and health outcomes at local and national level. This study focuses only on access to insurance. However, in reality, the uptake of the insurance was just a starting point towards health of migrants. There are many more factors involved, including availability of health services, affordability of a patient, quality of care, and relationship between providers and service users. Besides, the policy recommendations above are just a rough suggestion based on academic evidence. In practice, to translate these recommendations into action, there should be further studies on the implementation feasibility. These include a study to prioritise which policy should be implemented first and which should be implemented later. This means the suggested study should incorporate all needed information, including resistance (and/or acceptance) of the wider Thai public, the cost of the policy, windows of opportunities, and the political atmosphere at a particular period.

## 5. Conclusions

With the GMB and the CLD, it is clear that the access to public health insurance among migrants in Thailand is a complex matter. The existence of insurance itself is not a standalone measure. In fact, the issue is deeply involved and interplays with the concept of economic necessity and national security. It is inextricably linked with many other measures that are not the mandate of the health sector, particularly the NV process and the issuance of a work permit. The HICS and SSS are the key health insurance schemes for migrants in Thailand. Both are publicly run but are different in payment mechanism designs and target beneficiaries. The SSS focuses more on formal-sector employees while the HICS targets the informal-sector employees. Access to health insurance is beneficial for financial protection among the insurees and at the same time partly helps improve the financial status of health facilities. Challenges that hinder access to insurance include poor law enforcement on the employers of migrants, inadequate awareness of the existence of the insurance among migrants, and unaffordability of the insurance premium. The leverage points that potentially contribute to the enrollment in the HICS and SSS are the NV and the issuance of work permits. The government should be the key player in moving this issue forward while harnessing efforts from diverse social partners. According to this idea, all undocumented migrants should be enforced legally to undertake NV and be insured with the public insurance schemes. Strict law enforcement is key. Support from the government and social partners for migrants who are unable to afford the insurance price should be in place. To address the precarious legal status of migrants, the effort of the Thai government alone cannot guarantee NV success, as the process also needs mutual cooperation with the governments from the migrants’ countries of origin.

## Figures and Tables

**Figure 1 ijerph-17-05113-f001:**
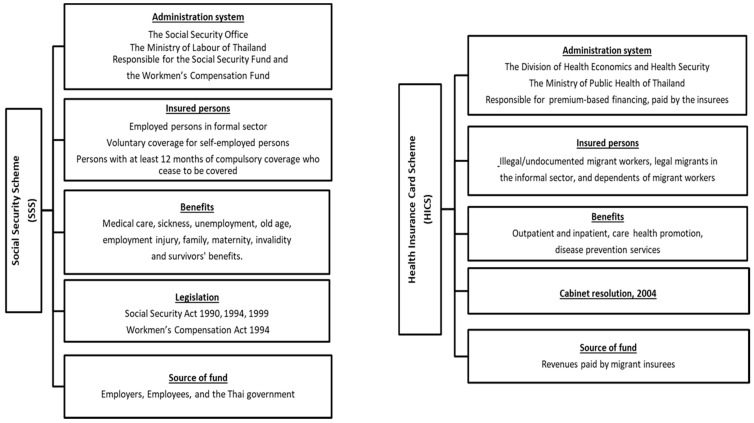
Characteristics of the SSS and HICS.

**Figure 2 ijerph-17-05113-f002:**
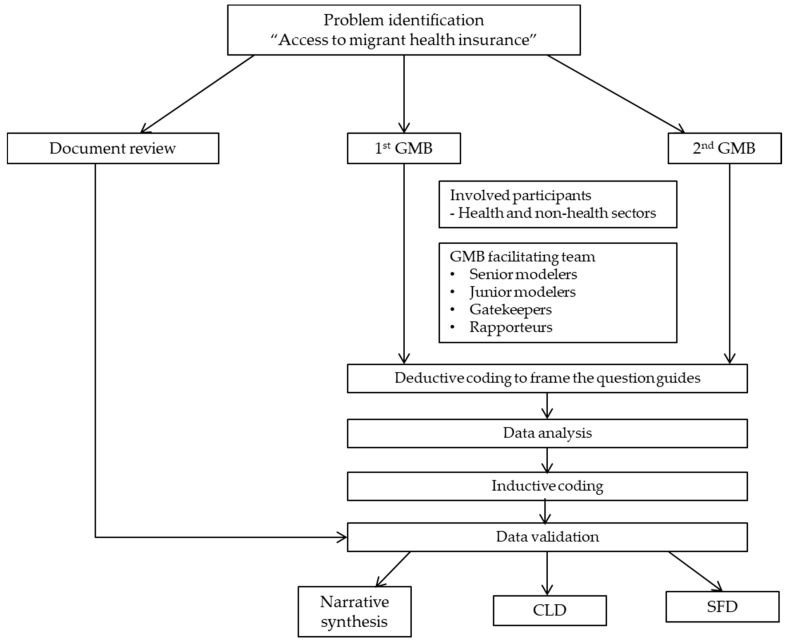
The process of data analysis.

**Figure 3 ijerph-17-05113-f003:**
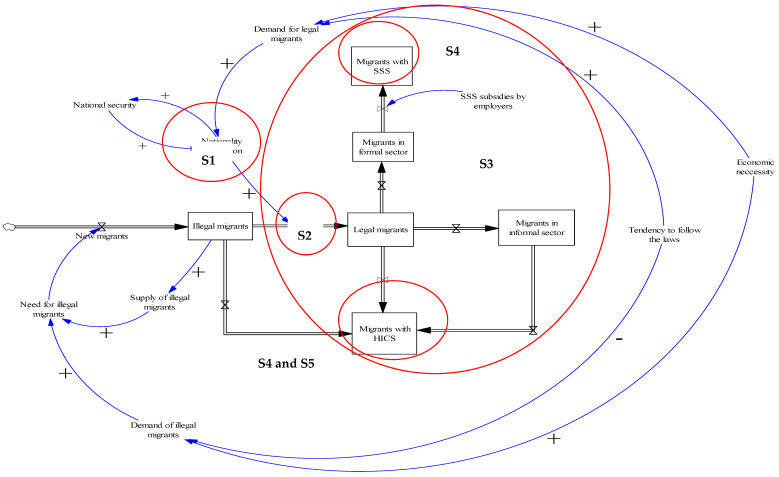
The SFD of the dynamics of the legalisation process and access to health insurance (SSS and HICS) by migrant workers. Note: Subtheme 1: NV process as the most critical step for legalising the precarious status of undocumented migrants. Subtheme 2: Role of private hospitals in health examination before enrolling in the insurance. Subtheme 3: Interim period between enrollment in the SSS and effective activation of insurance. Subtheme 4: Practical problems originating from the difference in the design between the SSS and the HICS. Subtheme 5: Data recording system of the HICS. HICS = Health Insurance Card Scheme; SSS = Social Security Scheme. (+) positive influence. (-) negative influence.

**Figure 4 ijerph-17-05113-f004:**
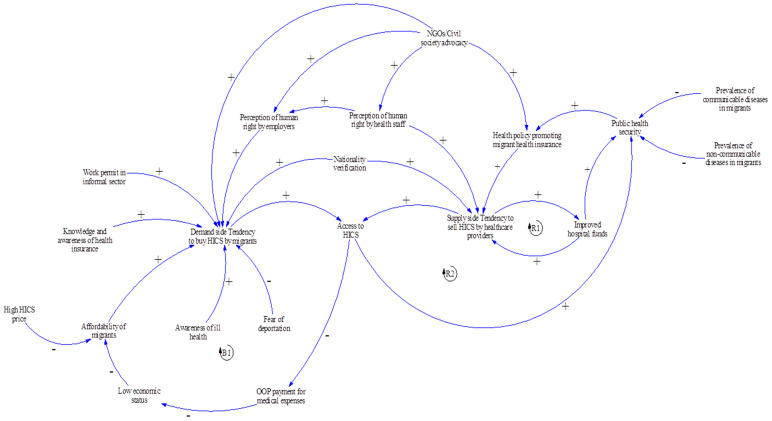
The CLD of the dynamics of access to health insurance (Health Insurance Card Scheme (HICS)) by migrant workers. Note: OOP = Out-of-pocket. (+) positive influence (-) negative influence.

**Table 1 ijerph-17-05113-t001:** Socio-demographic characteristics of stakeholders engaging in the 1st and 2nd GMB.

Variable	Category	*n* (Percentage)
Experience in the field of migrants’ health	<10	10 (50)
	10–19	2 (10)
	20–29	2 (10)
	≥30	6 (30)
Career level	Coordinator	2 (10)
	Middle	8 (40)
	Senior	10 (50)
Organization	Government authority	-
	Health sector	10 (50)
	Non-health sector	1 (5)
	Civil society	8 (40)
	Academic sector	1 (5)
Total	-	20 (100)

**Table 2 ijerph-17-05113-t002:** The health insurance schemes for migrants and dependents.

Domain	Social Security Scheme (SSS)	Health Insurance Card Scheme (HICS)
Financing mechanism	Tri-partite contribution 5% of employees’ income with subsidies of employers at 5% and Thai government at 2.75%	Premium-based financing HICS revenues pooled at the central MOPH and then decentralised to the local health facilities
Coverage duration	As long as the tri-partite contribution still continues	1. Migrant aged 18 years and over THB 2100 for 1 year THB 1400 for 6 months THB 1000 for 3 months 2. Dependents aged 7 years and over but not more than 18 years THB 2100 for 1 year THB 1400 for 6 months THB 1000 for 3 months 3. Dependents aged not over than 7 years THB 730 for 2 years THB 365 for 1 year
Contract facilities	Almost all public hospitals and some contracted private hospitals	Almost all public hospitals; no contracted private hospitals
Health benefit package	Outpatient, inpatient, accident and emergency, high-cost care	Outpatient, inpatient, accident and emergency, high-cost care but excluding renal replacement therapy and treatment for psychosis and drug dependence

Source: Modified from Suphanchaimat et al. [9] and Division of Health Economics and Health security, Ministry of Public Health, Thailand (2019) [29].

**Table 3 ijerph-17-05113-t003:** Summary of key findings of the subthemes.

Subtheme (S)	Key Findings
S1 NV process as the most critical step for legalising the precarious status of undocumented migrants	Unclear policy message about the implementation of a pink card (a temporary identity card for migrant workers). Migrants who passed the NV would acquire temporary passport and valid visa.
S2 Role of private hospitals in health examination before enrolling in the insurance	Aggravated during transition of job types when a migrant changed jobs from the formal to informal sector. Fear of being caught by police hindered the access to government registration among undocumented migrants.
S3 Interim period between enrollment in the SSS and effective activation of insurance	Unfair practices of selling HICS with inappropriate period of coverage during the interim period of SSS enrollment. Possibility to avoid health insurance enrollment after receiving health check from private hospitals.
S4 Practical problems originating from the difference in the design between the SSS and the HICS	Avoidance of health insurance due to difficulties in both SSS and HICS payment.
S5 Data recording system of the HICS	Unclear report of HICS selling by the central MOPH. Avoidance of report for high-cost treatment by health facilities.

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
