# Peer review of "Understanding the Problem of Access to Public Health Insurance Schemes among Cross-Border Migrants in Thailand through Systems Thinking"

_ijerph, 2020, doi:10.3390/ijerph17145113_

Round 1
Reviewer 1 Report
Abstract
Good, but no abbreviations here.
Introduction
The matter is very complex to understand. Please try to visualize the two main insurance schemes with pictures or a pictogram.
P2: 9 abbreviations within one Paragraph (p 2). Is too much… please check, if some of the abbreviations are unnecessary or are not mentioned later in the paper.
Methods
Section 2.3 , p 3, line 120 and following: if possible, please make a flowchart or picture to show the process of data management.
Please ad topics about data security, data management and anonymization of the material.
Results
P 5 line 169 and following please also here delete some abbreviations.
P 6, line 183 and following: what is “WCMW” and “MCWM”?
Please present the results also in a table or graph for better understanding of the sub-themes etc.
Discussion
Please start the discussion-section with a very brief summary of the main findings and the type of study/design
Please cite other literature from your own and other countries and delete the repetitions of results, e.g. on p. 11, line 347 and following and p 12, line 376 and following.
Try to shorten the whole discussion, it contains too much (redundant) results, but too little reflection: which findings can be shared with other health systems or countries? Please end the discussion with more questions and suggestions about further research and ideas.
Table 2 (p. 7) is too detailed (“coverage duration”) please try to shorten the whole table.
Please explain the abbreviations below figure 1
Author Response
- Good, but no abbreviations here (in the abstract).
- Thank you. I have deleted abbreviations in the abstract; please find it in the draft article.
- The matter is very complex to understand. Please try to visualize the two main insurance schemes with pictures or a pictogram.
- The details of the two health insurance schemes were shown briefly in Table 2.
- Abbreviations within one Paragraph (p 2). Is too much… please check, if some of the abbreviations are unnecessary or are not mentioned later in the paper.
- Thank you. We have checked abbreviation and deleted the unnecessary ones.
- Section 2.3 , p 3, line 120 and following: if possible, please make a flowchart or picture to show the process of data management.
- Please kindly find the chart of data management in Figure 1.
- Please ad topics about data security, data management and anonymization of the material.
- We have added more info about this point, please find it in line 138-149.
- P 5 line 169 and following please also here delete some abbreviations.
- The description of each abbreviation is summarized in the abbreviation part (line 528-533 ).
- P 6, line 183 and following: what is “WCMW” and “MCWM”?
- We corrected the typo error on that page to MCWM.
- Please present the results also in a table or graph for better understanding of the sub-themes etc.
- Thank you for this point. We integrated sub-themes into the CLD and SFD to show interconnections and locations of each sub-theme in the system.
- Please start the discussion-section with a very brief summary of the main findings and the type of study/design
- Thank you for your comment. We revised the discussion regarding your suggestion with brief summary of our findings in the first paragraph of discussion (Line 339-348).
- Please cite other literature from your own and other countries and delete the repetitions of results, e.g. on p. 11, line 347 and following and p 12, line 376 and following.
- Thank you for your constructive comment. We revised this part by adding the new evidence to support our findings, for example in line 389-392, 417-422.
- Try to shorten the whole discussion, it contains too much (redundant) results, but too little reflection: which findings can be shared with other health systems or countries? Please end the discussion with more questions and suggestions about further research and ideas.
- Thank you for your comment. We revised the discussion part with additional information and shortened some sentences to make it easier to read.
- Table 2 (p. 7) is too detailed (“coverage duration”) please try to shorten the whole table.
- We shortened Table 2 regarding your comment.
- Please explain the abbreviations below figure 1
- We added the information to describe abbreviation below Figure 1.
Reviewer 2 Report
To Authors:
If you wanna be considered for publication on the journal, I would suggest reorganize the information differently, also dedicating more time to the presentation of Results and Discussion sections, which are a bit confusing.
Good luck!
Author Response
14. If you wanna be considered for publication on the journal, I would suggest reorganize the information differently, also dedicating more time to the presentation of Results and Discussion sections, which are a bit confusing.
- Thank you for your constructive comment. The result and discussion parts were shortened and revised by reducing some unnecessary words and reorganized the ideas with support of some additional domestic and international evidence.
Reviewer 3 Report
The study addresses a highly topical subject of international importance. However, the manuscript is highly descriptive and I have a number of concerns, particularly around the results and discussion sections, and overall conclusions drawn. Overall, I felt the article has some merit but requires a number of significant changes to improve its rigor and interest to the journal's readership.
Introduction
- I found it difficult to follow the various government agencies, their roles and functions and the many context-specific acronyms used throughout the study. A summary table that captures acronyms, definitions and brief descriptions may be helpful for the reader.
- The introduction should provide some research context for the study; have others used a systems thinking approach to explore migrant access to public health insurance schemes? What are the advantages of this approach? How might your study add to what is known? What is of interest and relevance to readers outside Thailand and CLM?
Materials and Methods
- Approximate dates should be provided for each round of GMB; how many were there, were they held days/weeks/months apart and over what time period?
- Did all participants participate in each round? Was facilitation consistent between rounds?
- The deductive thematic analysis should be described in more detail. How were codes developed, were they informed by literature of personal expertise? Was a previously described approach used or modified?
Results
- In table 1, it would be useful to categorize participants' organizations so they aligned with how participants are described in the text. For example, I was not sure how participants from NGOs or independent health system research agencies were categorized.
- There appeared to be a high reliance on quotes from only two participants, out of 20, to support themes. I suggest you identify participants by number and organisation, e.g. Participant 3, NGO or Participant 7, public health officer. Unless sex is likely to influence how a participant responds, I don't think this level of detail is necessary. Should the supporting quotes be sourced from only two participants, care should be taken to ensure these participants' views have not skewed or dominated the reported results.
- The Stock and Flow diagram and the Causal Loop diagram were not reported in the Results section. As these were developed with direct input from participants and their generation was described in the methods, these should be reported in the results. I also suggest that each diagram is displayed on a single landscape page as they are very difficult to read in their current layout.
Discussion/Conclusions
- A large part of the discussion was spent describing the Stock and Flow and Causal Loop diagrams, which really belong in the methods.
- There were areas where the authors' opinions appeared to lead the discussion or that discussed participant responses that were beyond those described in the results, e.g. lines 326-328, 342-343, 353-354.
- I thought there were a couple of additional study limitations worthy of discussion. The first was that the participant demographics indicated that half were of senior career level. Do the authors think the imbalance of power between senior and lower-level participants may have limited discussion in the GMBs, especially as facilitation was kept at a minimum? The other potential limitation was the use of a deductive approach for thematic analysis, rather than an inductive approach. How did authors ensure the thematic analysis was rigorous and a true representation of the data?
- I found the most interesting aspect of the article to be the systems science approach and how GMBs were used to explore migrants' access to public health insurance schemes. However, this was only mentioned briefly in the discussion... What did the authors find were the advantages and disadvantages of this approach, how does this differ from others' experiences in the literature and what recommendations do they have for others seeking to undertake similar studies.
- The main conclusion drawn appeared to be a declaration that the NV process should be legally mandatory and strictly enforced. This was the only mechanism discussed for improving insurance uptake and did not appear to be well supported by the data reported. Stronger links between the reported results, discussion and conclusion need to be made. Does the literature support a punitive approach for improving public health insurance coverage for migrants? How is this approach, over others, supported by the Stock and Flow and Causal Loop diagrams?
Formatting and language
- The article was clearly written but contained a number of spelling (largely due to UK vs US English) and grammatical errors.
- Language occasionally became a little informal, e.g. not that strict (line 208).
- There were some minor formatting errors, such as inconsistent line spacing, throughout the manuscript.
Author Response
15. The study addresses a highly topical subject of international importance. However, the manuscript is highly descriptive and I have a number of concerns, particularly around the results and discussion sections, and overall conclusions drawn. Overall, I felt the article has some merit but requires a number of significant changes to improve its rigor and interest to the journal's readership. I found it difficult to follow the various government agencies, their roles and functions and the many context-specific acronyms used throughout the study. A summary table that captures acronyms, definitions and brief descriptions may be helpful for the reader.
- Thank you for your constructive comment. We added an abbreviation description part at the end of this article, see line 528-533.
16. The introduction should provide some research context for the study. Have others used a systems thinking approach to explore migrant access to public health insurance schemes? What are the advantages of this approach? How might your study add to what is known?
- Thank you for your constructive comment. Systems thinking and system dynamic modeling have been widely applied in the public health field for years, and some studies also used them as a tool to unpack complexity of migrant health policy in various ways. We believe that the benefits of these approaches can explore more structural mechanisms and context of access to migrant health insurance rather than individual factors arisen from migrants, employers or health providers themselves. Therefore, I added some evidence responding to this question in the introduction, methods, and discussion part, for example, please see line 78-93, 103-117, and 524-526. . However, in Thailand, it seems quite new to apply this approach in migrant health research. As such, this study tried to maximize this innovative thinking to explore important variables and their relationships, and then synthesized them into the stock and flow diagram as well as the causal loop diagram.
17. What is of interest and relevance to readers outside Thailand and CLM?
- This question is linked to Q16. We expect that the findings can provide lessons learned for neighboring countries of Thailand. Although the discussion provided further actions and recommendations mainly at a national level, this provides some evidence to link with migrant health policy at an international level, particularly in the Southeast Asian countries, please see line 423-431.
18. Approximate dates should be provided for each round of GMB; how many were there, were they held days/weeks/months apart and over what time period?
- There were two one-day GMBs run separately in July and August 2019. It took around three hours and engaged participants from diverse sectors. The GMB details were described in the methodology part, see line 126-133.
19. Did all participants participate in each round? Was facilitation consistent between rounds?
- Some participants joined both GMBs and some did not. However, the facilitation was conducted with the same protocol for both GMBs with the same group of facilitators.
20.The deductive thematic analysis should be described in more detail. How were codes developed?
- Thank you for your suggestion. We admit that the description of the analysis methods was not clear in its current form. Actually, we also applied a mix of inductive and deductive thematic coding during the analysis process with a content validation by feedbacks from experts in the field and stakeholders involving in both GMBs. Please see the revised text in line 158-173.
21. Were they informed by literature of personal expertise? Was a previously described approach used or modified?
- The data were mostly driven by participants’ experiences. Research team only informed them about the topic of discussion and then allowed participants to share their ideas about the process and challenges in access to migrant health insurance.
22. In table 1, it would be useful to categorize participants' organizations so they aligned with how participants are described in the text. For example, I was not sure how participants from NGOs or independent health system research agencies were categorized.
- We added a table describing the characteristics of each participant in the additional file to describe them in more details.
23. There appeared to be a high reliance on quotes from only two participants, out of 20, to support themes. I suggest you identify participants by number and organisation, e.g. Participant 3, NGO or Participant 7, public health officer. Unless sex is likely to influence how a participant responds, I don't think this level of detail is necessary. Should the supporting quotes be sourced from only two participants, care should be taken to ensure these participants' views have not skewed or dominated the reported results.
- Thank you for your comment. For each quote, I added the ID of each participant to show the distribution of participants’ opinion, for example, line 278, 290, 301.
24. The Stock and Flow diagram and the Causal Loop diagram were not reported in the Results section. As these were developed with direct input from participants and their generation was described in the methods, these should be reported in the results.
- Thank you for your comment. However, we would like to add SFD and CLD in the discussion part as the CLD was the synthesis of both findings from the GMB and perspectives of researchers which are supported by literature. That is, SFD and CLD are not the raw product of the group building process. We therefore considered that it would be better to place it in the discussion part. However, we integrated subthemes (which were shown in the Results) into SFD and CLD and please kindly find it in revised figures (please see Figure 2 and 3).
25. I also suggest that each diagram is displayed on a single landscape page as they are very difficult to read in their current layout.
- Thank you. We rearranged the figures to make it easier to read as suggested .
26. A large part of the discussion was spent describing the Stock and Flow and Causal Loop diagrams, which really belong in the methods.
- Same response as comment #24.
27. There were areas where the authors' opinions appeared to lead the discussion or that discussed participant responses that were beyond those described in the results, e.g. lines 326-328, 342-343, 353-354.
- Thank you for your comment. The discussion data were mainly based on participants’ opinions stemming from their own experience on this field. However, we also triangulated the discussion with reviewed documents, other interviews, and researcher’s observations. Therefore, we added some references as a source of the information. Please kindly find it in the revised document, line 355-362.
28. I thought there were a couple of additional study limitations worthy of discussion. The first was that the participant demographics indicated that half were of senior career level. Do the authors think the imbalance of power between senior and lower-level participants may have limited discussion in the GMBs, especially as facilitation was kept at a minimum?
- We agree with this comment. However, in this study we aimed to explore structural mechanisms of insurance access. Thus, perspectives from senior level officers are necessary. These participants are not super high-level policy makers; actually there are senior level officers that have policy linkage. However, we admit that this point could be a limitation though we deem that it does not devalue the findings. Further research is needed to explore more opinions from front-line workers or officers at local level. We have mentioned this point in line 439-443.
29. The other potential limitation was the use of a deductive approach for thematic analysis, rather than an inductive approach.
- Same response as comment #20.
30. How did authors ensure the thematic analysis was rigorous and a true representation of the data?
- We triangulated the results with reviewed literature, face-to-face interviews, and circulating draft report to all participants for feedback. More information on this process was described in the methodology part, line 158-173.
31. I found the most interesting aspect of the article to be the systems science approach and how GMBs were used to explore migrants' access to public health insurance schemes. However, this was only mentioned briefly in the discussion... What did the authors find were the advantages and disadvantages of this approach, how does this differ from others' experiences in the literature and what recommendations do they have for others seeking to undertake similar studies.
- Thank you for your comment. This question is linked to Q16 and 17.
32. The main conclusion drawn appeared to be a declaration that the NV process should be legally mandatory and strictly enforced. This was the only mechanism discussed for improving insurance uptake and did not appear to be well supported by the data reported. Stronger links between the reported results, discussion and conclusion need to be made. Does the literature support a punitive approach for improving public health insurance coverage for migrants? How is this approach, over others, supported by the Stock and Flow and Causal Loop diagrams?
- Thank you for your comment. In the CLD and SFD, the findings showed that the NV is central to public health security, national security and economic necessity. To highlight the impact of the NV on migrant health policy, we added more information on this issue and please kindly find it in the discussion part; line 355-362.
33. The article was clearly written but contained a number of spelling (largely due to UK vs US English) and grammatical errors.
- Thank you for your comment. We revised this draft article and improved the spelling and grammatical errors. A native English speaker was also asked to help edit this manuscript.
34. Language occasionally became a little informal, e.g. not that strict (line 208).
- This point was revised regarding your suggestion.
35. There were some minor formatting errors, such as inconsistent line spacing, throughout the manuscript.
- Thank you for your comment. We will work with the editor to check the format of this manuscript.
Round 2
Reviewer 1 Report
I do not understand the abbreviations in the comments of the authors e.g. “Q3” and following. Dies this refer to a letter for the reviewers, which is not available?
Introduction
The matter is very complex to understand. Please try to visualize the two main insurance schemes with pictures or a pictogram.
This has not been realized in the introduction.
Methods
Better now, but still too long
Results
Please present the results also in a table or graph for better understanding of the sub-themes etc.
Table 1 : same table as in the first version!
Table 2: too detailed (see my first review:” is too detailed (“coverage duration”) please try to shorten the whole table.)
Discussion
The discussion part still needs to be improved, see my first review:
“Try to shorten the whole discussion, it contains too much (redundant) results, but too little reflection: which findings can be shared with other health systems or countries? Please end the discussion with more questions and suggestions about further research and ideas.”
Has not been implemented, is now 100 lines longer than before.
To little reflexion, too much repetition of results (e.g. p. 13 line 443 and following: belongs to the result-section).
Author Response
- I do not understand the abbreviations in the comments of the authors e.g. “Q3” and following. Does this refer to a letter for the reviewers, which is not available?
- Thank you for your comment. Q3 referred to “question 3” raised by the reviewer.
- Introduction: The matter is very complex to understand. Please try to visualize the two main insurance schemes with pictures or a pictogram. This has not been realized in the introduction.
- Thank you for your comment. I have created a chart to describe main characteristics of each insurance scheme, please kindly find it in Figure 1.
- Method: Better now, but still too long
- I have reduced redundant words to keep it more concise. Please see the main text file.
- Please present the results also in a table or graph for better understanding of the sub-themes etc.
- Thank you for your comment. Table 3 was created following your suggestion.
- Table 1 : same table as in the first version!
- Thank you. I have shortened the table by deleting some unnecessary variables and please kindly find in in revised Table 1.
- Table 2: too detailed (see my first review:” is too detailed (“coverage duration”) please try to shorten the whole table.)
- Thank you for your comment. I have shortened the table to keep only key messages in that.
- The discussion part still needs to be improved, see my first review: “Try to shorten the whole discussion, it contains too much (redundant) results, but too little reflection: which findings can be shared with other health systems or countries? Please end the discussion with more questions and suggestions about further research and ideas.” Has not been implemented, is now 100 lines longer than before.
- I have revised the discussion part by reducing its length and added more information about suggestion for future research, please see line 529-540.
- To little reflexion, too much repetition of results (e.g. p. 13 line 443 and following: belongs to the result-section).
- Thank you for your constructive comment. I have improved the discussion with more details about reflexivity, please see line 518-520.
Reviewer 3 Report
Many thanks to the authors for the time and effort involved in resubmitting a much-improved manuscript.
Most of my concerns have been adequately addressed, with one or two exceptions.
The first is that I still think that key results are reported in the Discussion rather than the Results. The causal loop and stock and flow diagrams should be reported in the results section. A quick review of the studies cited by the authors indicated that these diagrams are generally reported in either the Results section or a combined Results/Discussion section. The problem with reporting the causal loop and stock and flow diagrams (which are direct outputs from the GMBs) in the discussion is that it becomes difficult for the audience to discern study results from reflections, recommendations and arguments. The Editor may want to make a recommendation about how best to structure the results and discussion sections.
The second is that some of the arguments/conclusions drawn don't appear to be well-supported by the evidence presented. In particular:
(lines 444-446) The government, in collaboration with all partners in society, should play an active role in encouraging all migrants to access the NV and obtain a work permit. Accordingly, strict law enforcement is vital. Employers who leave their migrant employees uninsured should be penalised.
(lines 555- 556) The employers who leave their migrant employees uninsured should be penalised.
It's unclear if this is an opinion voiced at the GMBs or the authors' recommendation. If an opinion voiced by GMB participants, the success (or not) of such a strategy should be discussed with reference to the literature. Similarly, if a recommendation by the authors, there should be some discussion about why this particular strategy is recommended over others. How do the literature and data support a punitive approach to improving migrant workers' access to the NV? This is especially important because employers and migrants did not participate in the study. There may be structural, cultural and/or process barriers to NV access that may not be addressed by a punitive approach.
Author Response
- Many thanks to the authors for the time and effort involved in resubmitting a much-improved manuscript.
- Thank you. We have revised the manuscript to make it more concise and constructive.
- Most of my concerns have been adequately addressed, with one or two exceptions. The first is that I still think that key results are reported in the Discussion rather than the Results. The causal loop and stock and flow diagrams should be reported in the results section. A quick review of the studies cited by the authors indicated that these diagrams are generally reported in either the Results section or a combined Results/Discussion section. The problem with reporting the causal loop and stock and flow diagrams (which are direct outputs from the GMBs) in the discussion is that it becomes difficult for the audience to discern study results from reflections, recommendations and arguments. The Editor may want to make a recommendation about how best to structure the results and discussion sections.
- Thank you for your suggestion. I have removed SFD and CLD to the result part. And the discussion was now reduced in length. However I have added some more reflections and policy implications.
- However, I rearranged the order of the themes. “Role of private hospitals in health examination before enrolling in the insurance” became theme 3.2.2, and “Interim period between enrolment in the SSS and effective activation of insurance” became theme 3.2.3.
- The second is that some of the arguments/conclusions drawn don't appear to be well-supported by the evidence presented. In particular: (lines 444-446) The government, in collaboration with all partners in society, should play an active role in encouraging all migrants to access the NV and obtain a work permit. Accordingly, strict law enforcement is vital. Employers who leave their migrant employees uninsured should be penalised, (lines 555- 556). The employers who leave their migrant employees uninsured should be penalised.
- Thank you for your constructive comment. I have diluted the tone of this message. Please kindly find it in the discussion, 505-520.
- It's unclear if this is an opinion voiced at the GMBs or the authors' recommendation. If an opinion voiced by GMB participants, the success (or not) of such a strategy should be discussed with reference to the literature. Similarly, if a recommendation by the authors, there should be some discussion about why this particular strategy is recommended over others. How do the literature and data support a punitive approach to improving migrant workers' access to the NV? This is especially important because employers and migrants did not participate in the study. There may be structural, cultural and/or process barriers to NV access that may not be addressed by a punitive approach.
- Thank you for your comment. I have revised this part by linking more findings to the discussion. The message about punitive measure was toned down.